# Contrasting responses of summer precipitation to orbital forcing in Japan and China over the past 450 kyr

Taiga Matsushita[1], Mariko Harada[2*], Hiroaki Ueda[3], Takeshi Nakagawa[4], Yoshimi Kubota[5], Yoshiaki Suzuki[6], Youichi Kamae[3]

[1]Graduate School of Science and Technology, University of Tsukuba, Tsukuba, Ibaraki 305-8577, Japan
[2]Department of Earth and Planetary Sciences, Tokyo Institute of Technology, Tokyo 152-8550, Japan
[3]Faculty of Life and Environmental Sciences, University of Tsukuba, Tsukuba, Ibaraki 305-8577, Japan
[4]Research Centre for Palaeoclimatology, Ritsumeikan University, Kusatsu, Fukui 525-8577, Japan
[5]Department of Geology and Paleontology, National Museum of Nature and Sciences, Tsukuba, Ibaraki 305-0005, Japan
[6]Marine Geology Research Group, National Institute of Advanced Industrial Science and Technology, Tsukuba, Ibaraki 305-8567, Japan

*Correspondence to*: Mariko Harada (harada.m.an@m.titech.ac.jp)

**Abstract.** Understanding orbital-scale changes in East Asian summer monsoon (EASM) precipitation is a fundamental issue in paleo-climate research for assessing the response of the East Asian monsoon to different climatic forcings, such as insolation, ice volume, and greenhouse gases. However, owing to the inconsistencies between different proxies, the fundamental driving force for EASM variability remains controversial. The present study simulated the global climate under the given insolation changes over the past 450 kyr using a climate model, Meteorological Research Institute Coupled General Circulation Model version 2.3 (MRI‑CGCM2.3). Changes in summer insolation over East Asia resulted in distinct climatic responses in China and Japan: an increase in summer insolation led to increased summer precipitation in China and decreased summer precipitation over Japan. Composite analyses of simulated climate under strong boreal summer insolation suggest that the warming of the Indian Ocean occurs under intense insolation, resulting in the intensification of the North Pacific subtropical high (sub-high). The northern shift of the monsoon front, associated with the intensified sub-high, leads to an increase in rainfall in the coastal area of China. In contrast, the intensity of the EASM around Japan is affected by the strength of the North Pacific High. Under strong insolation, the increase in thermal contrast between the North American continent and the North Pacific Ocean intensifies the North Pacific High, decreasing the summer precipitation around Japan. Thus, strong regional differences in the effects of solar insolation variability on summer precipitation in East Asia exist due to interactions with different ocean basins.

# 1 Introduction

The East Asian summer monsoon (EASM) plays an essential role in the hydrological cycle over East Asia, affecting food production, water supply, and other extreme events such as floods and droughts in densely populated areas (An et al., 2015). Understanding the variability of the EASM is crucial in assessing future climate projections (e.g., global warming) in East Asia and, eventually, in human social and economic activity. Notably, investigating the long-term variation in the EASM over geological time is of fundamental importance in modern and paleo-climatology, as it enables us to reveal the mechanisms of the EASM variability under different climatic forcings.

The late Pleistocene climate, characterised by periodic changes in the Northern Hemisphere summer insolation and the high-latitude ice volume, has received considerable attention. The insolation cycle is caused by variations in three of the Earth's orbital parameters: precession (23-kyr periodicity), obliquity (41 kyr), and eccentricity (100 kyr) (Berger, 1978; Berger and Loutre, 1991). Although changes in eccentricity have a relatively small effect on insolation variability, the eccentricity cycle is considered an underlying cause of the 100 kyr glacial–interglacial cycle (Lisiecki and Raymo, 2005) through interactions between the climate, ice sheets, and lithosphere–asthenosphere system (Abe-Ouchi et al., 2013). These orbital and glacial factors might have been the major driving forces of EASM variability during the late Pleistocene (Wang et al., 2017).

Geological records indicate that the EASM is affected by changes in insolation, ice volume, or both (Cheng et al., 2016; Clemens et al., 2018; G. Liu et al., 2020; Sun et al., 2006, 2015, 2019). For example, loess–paleosol records from the Chinese Loess Plateau, Northern China, have displayed the coexistence of 100-, 41-, and 23-kyr periods, with a distinguished 100-kyr periodicity associated with the glacial–interglacial cycle (Beck et al., 2018; Nie et al., 2017; Sun et al., 2015). The local seawater $\delta^{18}O$ of the East China Sea has been dominated by 100-kyr and 41-kyr cycles with almost no precession variance (Clemens et al., 2018). These results suggest that the EASM is more sensitive to high-latitude ice volume variations than to direct insolation changes. In contrast, speleothem $\delta^{18}O$ records from caves in southern China showed an almost pure precession cycle, implying the dominance of direct insolation forcing in summer precipitation over southern China (Cheng et al., 2016; Wang et al., 2008; Zhang et al., 2019). Similarly, a long pollen record from Lake Biwa in Japan contains distinct precession and obliquity signals, suggesting that the EASM has been controlled primarily by insolation forcing and that the effect of ice volume changes becomes predominant only when insolation is weak (Nakagawa et al., 2008).

The different responses of these proxies are probably interpreted as regional heterogeneity in the relative impact of EASM driving forces. For instance, G. Liu et al. (2020) suggested that the lack of glacial–interglacial cycles in the speleothem $\delta^{18}O$ may be explained by the moisture transport pathway effect, in which glacial–interglacial cycles in the monsoon moisture source decrease landward. However, due to proxy data being scattered in limited areas, forming a spatially continuous interpretation has proven difficult. In addition, proxies potentially have complex sensitivity to several climatic factors (e.g., temperature changes and moisture sources), causing difficulty in the interpretation of records (An et al., 2015; Clemens et al., 2010, 2018; X. Liu et al., 2020).

To reveal the mechanisms of EASM variability, numerous modelling studies with different boundary conditions have been conducted over the last decades (e.g., Braconnot et al., 2007a; Chen et al., 2011; Dai et al., 2021; Kutzbach, 1981; Kutzbach et al., 2008; Li et al., 2013; Liu et al., 2014; Lyu et al., 2021; Shi et al., 2011; Suarez and Held, 1976; Sun et al., 2015, 2018, 2019, 2021, 2023; Thomas et al., 2016; Weber and Tuenter, 2011; Yin et al., 2008, 2009, 2014; Yin and Berger, 2012; Zhao et al., 2020). Among these, numerous studies have consistently indicated the presence of the 20-kyr precession cycle in summer precipitation in most EASM regions(Li et al., 2013; Sun et al., 2015; Lyu et al., 2021; Weber and Tuenter, 2011), suggesting that insolation appears to be the key factor determining the variability of EASM (Dai et al., 2021; Li et al., 2013; Sun et al., 2023). The impact of ice volume on summer precipitation varies between regions (Sun et al., 2015, 2021; Lyu et al., 2021); for instance, Lyu et al. (2021) indicated that the effect of the glacial–interglacial 100-kyr cycle becomes significant in the 20–25°N region, whereas in other regions, summer precipitation is primarily influenced by the 20-kyr precession cycle. These results suggest that, in the EASM regions, summer rainfall variability is affected by orbital and glacial forcings; however, the relative impact of each forcing differs among regions (e.g., Dai et al., 2021; Sun et al., 2015). The proposed mechanisms that cause the spacial and temporal variability of the EASM are often attributed to the anomalous western North Pacific subtropical high (sub-high) (Dai et al., 2021; Liu et al., 2014; Sun et al., 2015), which causes shifts in the seasonal Meiyu/Baiu rain bands (Yihui and Chan, 2005), and variation in southerly wind provides moisture into EASM regions (Liu et al., 2014). However, the simulated regionalities varied between studies owing to the unclear position of the sub-high depending on models (Dai et al., 2021), and the detailed mechanisms of the sub-high fluctuation remain ambiguous. In addition, determining the individual impact of insolation and ice volume changes on EASM variability remains challenging because of the complex relationship between orbital and glacial forcings (Yin et al., 2014; Lyu et al., 2021). Thus, monsoon variability under individual forcing (insolation or ice volume) should be analysed more substantially in terms of the global climate system, including atmospheric circulation, atmosphere–ocean interactions, and teleconnections.

In the present study, we aimed to reassess the relative impact of insolation changes due to orbital forcing on EASM variability. We conducted 450 kyr-long idealised, time-sliced simulations with a boundary condition of the Earth's three orbital parameters (Berger and Loutre, 1999) to reconstruct orbital-scale EASM variability using a global climate model. Using the time-slice method, the quasi-equilibrium climate state of each period was simulated at an interval of 5 kyr. This study reveals the possible mechanisms responsible for regionally different responses of EASM variability to insolation changes through data-model comparisons over East Asia. Section 2 describes the climate model and the experimental design (or datasets). Section 3 explains the model results and compares the numerical results with proxy data. Section 4 discusses the possible climate systems associated with EASM variability that are responsible for the known regionality of proxy data. In Section 5, we summarise and conclude the paper.

## 2 Model and experimental design

### 2.1 Model

Experiments were performed using the Meteorological Research Institute Coupled General Circulation Model version 2.3 (MRI‑CGCM2.3) (Yukimoto et al., 2006). The atmospheric component of the model was a spectral model with T42 spatial resolution (approximately 2.8°) and 30 vertical layers, which was based on Shibata et al. (1999). The oceanic component of the model was a Bryan–Cox-type ocean general circulation model with a horizontal resolution of 2.5° longitude and 2.0–0.5° latitude and 23 vertical layers. Details of the model are described in Yukimoto et al. (2006).

The land component, incorporating vegetation effects, relies on the Simple Biosphere (SiB) model (Sellers et al., 1986; Sato et al., 1989). In this model, ice sheets are treated as one of the vegetation types which is reflected in the albedo in the simulation. Vegetation is fixed as a boundary condition in each simulation. Thus, the dynamics of vegetation and ice sheets are not integrated; adjustments in ice volume can be accommodated by modifying land vegetation and altitude, specified as boundary conditions. Sea ice compactness and thickness are forecasted utilising thermodynamics, horizontal advection, and diffusion principles.

MRI‑CGCM has been widely used for future climate prediction (e.g., Kitoh, 2007; Ueda et al., 2006) and paleo-climate simulations (e.g., Braconnot et al., 2007a; Kamae et al., 2016, 2017; Kitoh et al., 2007; Ueda et al., 2011). The reproducibility of MRI‑CGCM2.3 has been studied by Yukimoto et al. (2006) and indicates good agreement with observations in the distribution of surface air temperature and precipitation. The model effectively simulates summer precipitation over East Asia, including its onset timing and migration (Yukimoto et al., 2006), and has been used for paleo-Asian monsoon simulations, including the Last Glacial Maximum (LGM) (Ueda et al., 2011), the Medieval Warm Period, and Little Ice Age (Kamae et al., 2017).

### 2.2 Experimental design

We used a time-slice approach by dividing the last 450 kyr into 91 periods (5 kyr intervals). The quasi-equilibrium climate state was evaluated during each period. While this approach may not fully capture the transient behaviour of climate systems, as demonstrated in recent transient simulations (Chen et al., 2011; Clemens et al., 2018; Kutzbach et al., 2008; Timmermann et al., 2007), the mean climate states obtained through the time-slice method are well-suited for evaluating the responses of prescribed external forcings to monsoon variability. This methodology has been employed in numerous studies (Kutzbach, 1981; Liu et al., 2014; Wen et al., 2016). The time-slice approach can calculate the long-term integral in the same period, enabling us to discuss the seasonal evolution of the EASM in addition to long-term climate responses to orbital forcing (Cubasch et al., 1995).

As a boundary condition, insolation was simulated from the Earth's three orbital parameters (longitude of perihelion as precession, obliquity, and eccentricity) at each corresponding time period based on Berger and Loutre (1999). In all time-sliced experiments, any other time-varying boundary conditions, such as vegetation, the concentration of greenhouse gases,

ice sheets, topography, and land–sea boundaries, were prescribed at the pre-industrial (PI; 0 ka) level. In particular, the concentration of greenhouse gases was set to the same value as that of the Paleo-climate Modelling Intercomparison Project 2 (PMIP2) (Braconnot et al., 2007a, 2007b). Although ice volume changes and variations in greenhouse gas concentrations might impact EASM variability (Clemens et al., 2018; Yin et al., 2008; Yin and Berger, 2012), we focused exclusively on the effect of orbital changes to evaluate the relative impact of insolation changes on EASM. All experiments were conducted without flux adjustments. To eliminate the effects of climate drift (Kitoh et al., 2007), the model was integrated for 230 years prior to the main experiments. Then, the result of the 230$^{th}$ year was forwarded to the main experiments as an initial condition. After integrating another 55 years into the main experiments, the average output of the last 50 years was analysed as the representative climate state of each time period.

The 'anomaly' in the climate state of each period, defined as the difference in climate state between PI and each time period, was analysed to capture the effect of insolation variability. In addition, we defined a strong period (SP) as the period in which the summer insolation anomaly at the top of the atmosphere in low-mid latitude (20° N–40° N) exceeded 1 standard deviation (SD). Seasonality was discussed based on the Northern Hemisphere seasons. Specifically, this section used 'summer' to refer to June–August (JJA). Unless otherwise noted, all correlation coefficients calculated in this study exceeded the 99.9 % confidence level using Pearson's t-test.

**3 Simulated temporal climate variability over the last 450 ka**

In this section, we describe the simulated climate variability and its predominant cycles in the three main EASM regions (South East China: SEC, Chinese Loess Plateau: CLP, Japan: JP), defined as SEC (24° N–33° N, 105° E–120° E), CLP (34° N–38° N, 100° E–110° E), and JP (30° N–40° N, 130° E–143° E), respectively (Fig. 1). Figure 2 shows the simulated temporal variation in summer precipitation, annual precipitation, and summer temperature over the last 450 kyr (Fig. 2a) and the periodicity of each parameter  (Fig. 2b). Summer precipitation variability in SEC and CLP had a strong similarity to summer insolation and local summer temperature change. Notably, the maxima of summer precipitation precisely coincided with the SPs of summer insolation (Fig. 2a; hatched area). Furthermore, we found a strong positive correlation between summer insolation and precipitation over SEC ($R = 0.94, p < 0.001$) and CLP ($R = 0.78, p < 0.001$) (Fig. 3). These results confirm the significance of the insolation effect on the variability of EASM in China. This has been further supported by previous simulations using different models, such as CCSM3 (Li et al., 2013) and HadCM3 (Lyu et al., 2021). All three regions (SEC, CLP, and JP) exhibited similar variabilities in simulated local JJA temperature, consistent with insolation changes: summer temperature increases with increasing summer insolation. In JP, the relationship between summer insolation and precipitation was relatively unclear; the minima of summer precipitation corresponded to certain summer insolation SPs (Fig. 2a). The correlation coefficient between summer insolation and precipitation in JP was negative ($R = -0.63, p < 0.001$) (Fig. 3). These results suggest that the effect of summer insolation on precipitation varies between regions, with notable differences in the response observed between China and Japan.

In all three regions, annual precipitation varied over a 23-kyr cycle; however, its amplitude was smaller than that of summer precipitation variability (Fig. 2a). This likely occurred due to the fluctuations in precipitation in the summer and other seasons canceling each other out. Moreover, the contribution of other seasonal precipitation to annual precipitation varied among regions. In SEC, summer variation in precipitation did not coincide with annual precipitation, suggesting that the precipitation in other seasons offset the influence of summer precipitation on the annual precipitation. In CLP and JP, annual precipitation fluctuated in phase with summer precipitation, which is consistent with the fact that the contribution of summer precipitation in the area is relatively larger than that of precipitation in other seasons.

The periodicity of change in summer precipitation in each region had different characteristics (Fig. 2b). In the SEC and CLP, summer precipitation was dominated by precession bands (19 and 23 kyr), whereas in JP, it was dominated by an obliquity band (41 kyr), and the precession bands were not as strong as in SEC and CLP. A slight 56-kyr band, corresponding to one of the periodicities of precession (Berger, 1978), was found in the CLP summer precipitation variability, though it was neither visible in insolation nor in other regions. A 100-kyr eccentricity band was not found in any region because the effect of eccentricity on insolation is generally minimal compared to that of precession and obliquity (Schneider et al., 2011).

We compared the model results with proxy data comprising the oxygen isotope ratio records ($\delta^{18}$O) of speleothems from Chinese caves (located in SEC) (Cheng et al., 2016), meteoric [10]Be preserved in Pleistocene Chinese loess (CLP) (Beck et al., 2018), and a pollen record from Lake Biwa (JP) (Nakagawa et al., 2008) (Fig. 2). The represented monsoon index differed between proxies. Speleothem $\delta^{18}$O is used to document summer monsoon variability (Wang et al., 2017), whereas CLP [10]Be records indicate the annual amount of precipitation, which can be treated as summer precipitation according to Beck et al. (2018). The pollen record in JP represents amounts from April to September (Nakagawa et al., 2008).

The model results were highly consistent with the cave $\delta^{18}$O records in SEC (Fig. 2a), with the correlation between simulated summer precipitation and $\delta^{18}$O records in SEC being $R = -0.56$ ($p < 0.001$). The power spectrum densities also exhibited good agreement between the model results and proxy data, in which only the precession cycle was distinguished in the SEC (Fig. 2b). Proxy records from Chinese caves may be affected by climatic factors beyond summer precipitation variability (e.g., Clemens et al., 2010; Wang et al., 2017). For instance, Clemens et al. (2010) proposed that the speleothem $\delta^{18}$O may reflect winter temperature variability. Nevertheless, our results indicate that changes in solar insolation alone have an impact on summer precipitation in China. The simulated variation in summer and annual rainfall in CLP was consistent with annual precipitation inferred from [10]Be data (Fig. 2a), although the correlation was not as strong as in SEC (with the correlation between simulated summer precipitation and the [10]Be record being $R = 0.42$, $p < 0.001$). We further compared the simulated climate with available proxies in CLP by comparing simulated mean annual air temperature (MAAT), EASM intensity (summer precipitation) and EAWM intensity (index based on Wang and Chen (2014)) with MAAT inferred from branched glycerol dialkyl glycerol tetraethers (brGDGTs), loess magnetic susceptibility as an EASM proxy, and mean grain size as an EAWM proxy, respectively (Thomas et al. 2016) (Fig. 4). The simulated variations in the EASM and EAWM were generally consistent with precession cycles appearing in the proxies (Fig. 2a and Fig. 4a); however, variation in 100-kyr periodicity, dominant in the proxy data, was not reproduced in the simulation (Fig. 2b and Fig. 4b). Relatively no variation

was observed in mean annual air temperature (MAAT) in the simulated results, because as with precipitation, the effects of solar radiation on temperature were offset on an annual basis. The deviation from the respective proxies for MAAT, EASM, and EAWM was likely derived from the lack of an effect elicited by 100-kyr ice sheet variability and related internal feedback in our model. These characteristics in SEC and CLP support the idea that summer precipitation variability in China is strongly influenced by changes in summer insolation, particularly in SEC. The inconsistency between proxy and model results in CLP indicates that, although insolation certainly affects summer precipitation variability in Northern China, the actual variability of summer precipitation should be assessed by incorporating other forcings and internal feedback, such as global ice volume change and variation in greenhouse gases.

The simulated summer rainfall variability in JP appeared inconsistent with the rainfall reconstructed from proxy data (Fig. 2a), with the correlation being negative and below a significant level ($R = -0.23, p = 0.01$). The similarity in the dominant cycles (dominance of precession and obliquity bands) between the model results and proxy data may support the idea that changes in summer insolation affect the monsoon variability in JP (Fig. 2b). However, in the simulation, the strong summer insolation caused a decrease in summer precipitation in JP, while the proxy appeared to indicate the opposite response. Based on the assumption that the change in land–sea thermal contrast and resultant EASM variability are negatively correlated with the annual temperature range (Tvar) (Nakagawa et al. 2008), we compared the simulated results with variation in Tvar and AMJJAS rainfall reconstructed from pollen records (Fig. 5). The results suggested that Tvar negatively correlated with summer precipitation; however, the simulation and proxies showed contrasting Tvar responses to insolation. In the simulated results, the increase in summer insolation caused the increase in Tvar, leading to a decrease in summer precipitation. The inconsistency between the model results and proxy data indicates that the summer rainfall in JP represented by pollen records cannot be fully interpreted based on the previously proposed insolation and Tvar effects; it contains the effect of complex climatic forcings and internal feedback. The relationship between insolation and Tvar, as well as the summer precipitation variability in JP should be verified using multiple proxies and models in future studies.

## 4 Possible climate dynamics responsible for EASM variability

### 4.1 Mechanism of variability in South East China and Chinese Loess Plateau in-phase with insolation

Figure 6a depicts the composite distribution of summer precipitation anomalies in the SPs, along with anomalies in sea level pressure (SLP) and winds. For comparison, the simulation results for PI (0ka) are shown in Fig S1 in the Supplement. Additionally, the global distribution of the anomalies is presented in Fig. S2. The increase in precipitation was observed over China, and the trend was more pronounced in SEC (Fig. 6a). This is consistent with spatial heterogeneity reported in Sun et al. (2015), where insolation forcing had a more significant impact on low-latitude and coastal areas in China. It has been suggested that sub-high variability played an important role in past monsoon variability over East Asia (Dai et al., 2021; Liu et al., 2014; Sun et al., 2015), though the position of the sub-high and its role on EASM rainfall depends on models. In our results, the

anomalously high pressure was noticeable over the Philippine seas, which intensifies southerly winds at their western edge (Fig. 6a and Fig. S2a). Therefore, we suggest that, under intense solar insolation, the active moisture transport resulting from the strengthened southerly wind at the western edge of the sub-high led to increased summer precipitation over China. This effect would be more significant in SEC than in CLP. While our simulation did not include a change in ice volume, the incorporation of ice sheets will cause a further decrease in summer precipitation in CLP during the glacial periods (Sun et al., 2015), as global cooling and a reduced land–sea pressure gradient contrast weaken the EASM circulation (Lyu et al., 2021).

Figure 6b depicts the distribution of the sea surface temperature (SST) and surface wind anomalies in SPs. SST increased over almost the entire Indian Ocean except for the central equational region (Fig. 6b and Fig. S2b), corresponding to increasing summer precipitation over the Indian Ocean (Fig. 6a). In the modern climate, the sub-high is considered to be formed and maintained through air–sea interaction over the Indo and western Pacific Ocean (Xie et al., 2016). The warming of the Indian Ocean excites the Matsuno–Gill response (Gill, 1980; Matsuno, 1966) through tropospheric Kelvin wave propagation, forcing the anomalous anticyclone over the South China Sea and the Philippine Sea, the sub-high in summer (Xie et al., 2009). The notable consistency of basin-wide warming of the Indian Ocean and anomalous easterly winds extended from the Indian Ocean to the western North Pacific around Philippines indicates that the same mechanisms can be applied to increasing EASM rainfall in SP. The tropospheric Kelvin wave propagation frequently occurs in SPs. Although the enhanced sub-high appears to be inconsistent with the warming South China Sea regarding local air-sea feedback (Fig. 6b), it may be explained that the effect of the Indian Ocean overwhelms the local SST feedback (Xie et al., 2009).

Xie et al. (2009) assumed that Indian Ocean warming occurs in summers following the waning of El Niño events. However, even in the absence of ENSO variability, the presence of negative SST anomalies in the tropical northwestern Pacific can lead to the formation and maintenance of sub-highs through wind evaporation SST (WES) feedback, the mechanisms of which is referred to as the Indo-western Pacific Ocean Capacitor (IPOC) mode (Xie et al. 2016). The IPOC mode involves not only the Kelvin response to Indian Ocean warming (Xie et al. 2009), but also the Rossby response to convective inactivity in the tropical northwestern Pacific Ocean. According to Wang et al. (2000), WES feedback operates as follows: SST cooling over the western North Pacific suppresses in situ convective heating, exciting a westward-propagating Rossby wave that forms an anomalous anticyclone in the Philippine Sea. The intensified northeast trade winds at the eastern edge of the anomalous anticyclone amplified the initial SST cooling via evaporation and wind stirring. Since background trade winds are essential for WES feedback, the contribution of WES feedback is significant from winter to spring, when trade winds prevail in the tropical northwest Pacific. Thus, the initial disturbances that form the IPOC mode (e.g., tropical Northwest Pacific SST lowering) are likely to occur in the spring season. To investigate whether WES feedback also occurred in the past, we verified the distribution of SST anomalies in spring (MAM) in the SPs. The SST cooling band extending from southwest to northeast and the in situ anomalous high were identified over the western North Pacific (Fig. 6c and Fig. S2c). The anomalous high near the Philippines, located slightly west of the negative SST region, likely corresponds to the Rossby response to the SST anomaly and intensifies the mean northeast trade winds at the eastern edge of the anomalous high in 10° N–20° N (Fig. 6c and Fig. S2c). Thus, WES feedback may have formed and maintained the sub-high in SPs, increasing the monsoon rainfall over the SEC and CLP. The

spring climate may reflect the effects of increasing seasonal variations in insolation (e.g., weak winter insolation), rather than strong summer insolation. This suggests that the EASM can be influenced not only by summer insolation but also by variations in insolation distribution across other seasons. Though the increase in SEC and CLP was attributed to the changes in SST, the causal mechanisms connecting solar insolation changes and SST anomalies remained unclear in this study, which would be an important future study.

## 4.2 Mechanism of anti-phase variability in Japan

The inverse phase between summer insolation and precipitation around JP was found in the model results (Fig. 3). Composite SLP anomalies over the North Pacific region (Fig. 6a) indicate that the salient anomalous North Pacific High (NPH), with a maximum at approximately 35°N 170°E was pronounced (~5 hPa) in SPs, the western edge of which overlaps around JP and reduces summer rainfall (Fig. 6a). In modern climatology, the variation in NPH is often attributed to the thermal contrast of diabatic heating between land and sea (Rodwell and Hoskins, 2001; Miyasaka and Nakamura, 2005). For example, the intensified thermal contrast between the North Pacific Ocean and the North American continent results in a stronger northerly wind over the western coast of the North American continent and cools SST by the enhanced coastal upwelling. As a result, the NPH may be intensified due to the parasol effect and/or radiative cooling because low-level clouds increase when the SST declines (Miyasaka and Nakamura, 2005). In the present study, the anomaly of vertically integrated diabatic heating in SPs was positive over the North American continent and negative over the North Pacific Ocean (Fig. 7a). Notably, the large sensible heating was found over land, contributing to the warming of the North American continent (Fig. 7b). In contrast, a negative anomaly in latent heat was observed over North Pacific Ocean (Fig. 7c). These features indicate the enhancement of thermal contrast in SPs, consistent with the intensification of NPH. The tropics, including the Maritime Continent, were characterised by remarkable diabatic heating anomalies (Fig. 7a). The anomaly derived mainly from latent heat (Fig. 7c), suggesting less convective activity over the South China and Philippine seas. The climate system discussed here may not reflect the actual variability because the model results did not reproduce the temporal variation suggested by proxies. Thus, the impact of additional forcing (ice sheets and greenhouse gases) on the precipitation in JP should be analysed in future studies.

## 5 Conclusions

The present study aimed to quantitatively assess the effects of insolation changes based on the three orbital parameters on EASM variability. The results showed that the increase in Northern Hemisphere insolation strongly intensifies the summer precipitation over China. This can be attributed to a climate system similar to the IPOC mode known in modern climatology. From spring to early summer in SPs, the WES feedback via negative SST anomalies in the tropical western North Pacific and the warming of the Indian Ocean in summer sustain the western Pacific subtropical high. The western Pacific subtropical high enhances the water vapour transport to the north, subsequently increasing the EASM rainfall in SEC and CLP. In contrast, the model results showed that the increase in insolation causes the suppression of the summer precipitation in JP

through the intensification of NPH. The model results coincide with the proxy records in the SEC region, whereas they were apparently inconsistent with proxy records in CLP and JP. This implies that insolation changes must have controlled the EASM rainfall in SEC. However, other external forcings, such as changes in ice volume and greenhouse gases, markedly influence the EASM variability in CLP and JP. Numerical experiments considering further external forcings, such as ice sheet volume and greenhouse gas fluctuations, are expected to clarify the actual EASM variability and its complex intrinsic system.

## Author contribution

TM, MH, HU, and TN designed this study. TM performed experiments. TM and MH wrote the manuscript. All the authors discussed the results and commented on the manuscript.

## Data availability

The output of climate simulations performed in this study (91 runs of global climate simulation) occupy several terabytes of data, and thus have not been made freely available. Nevertheless, they can be accessed upon request to the corresponding author. The area-averaged precipitation and temperature data used to create the time series and scatter plots are available as a Supplement to this paper.

## Competing interests

The authors declare that they have no conflict of interest.

## Acknowledgements

This work was supported by JSPS KAKENHI Grant Number 21H01197.

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

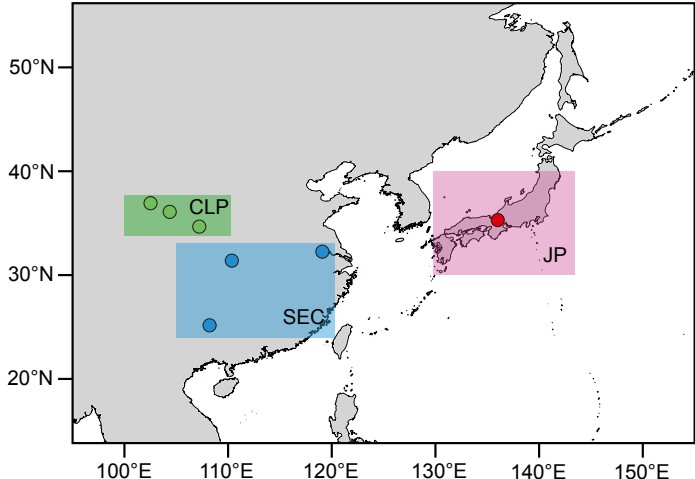

**Figure 1: Three EASM regions defined in this study.** Blue: South East China (24° N–33° N, 105° E–120° E), green: Chinese Loess Plateau (34° N–38° N, 100° E–110° E), red: Japan (30° N–40° N, 130° E–143° E). Circle plots indicate the locations of proxy records (Beck et al., 2018; Cheng et al., 2016; Nakagawa et al., 2008; Sun et al., 2015). The map was created using the free and open source QGIS.

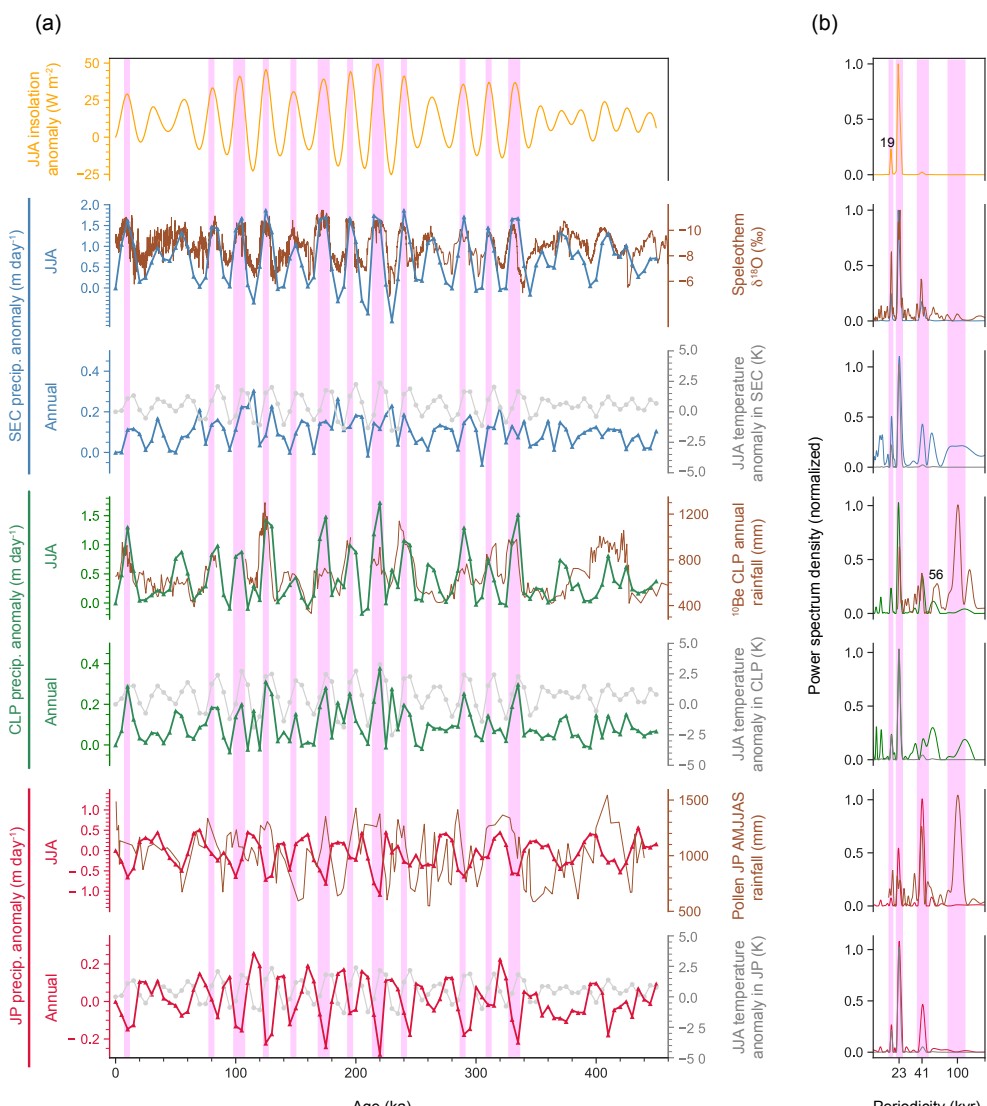

**Figure 2:** (a) Temporal variation in JJA insolation anomaly (orange, W m-2), simulated precipitation anomaly in SEC (blue), CLP (green), and JP (red). Insolation was averaged over 20° N–40° N. The precipitation anomaly unit is mm day[-1]. Brown lines indicate proxy records: Speleothem δ18O over SEC (‰, Cheng et al., 2016), 10Be-based annual rainfall over CLP (mm, Beck et al., 2018), and pollen-based AMJJAS rainfall variability in Lake Biwa in JP (K, Nakagawa et al., 2008). Grey lines indicate the simulated annual temperature anomaly (K) in each region. Variation in JJA insolation was simulated in the model using orbital parameters taken from Berger and Loutre (1999), and averaged over grid cells ranging from 20°N to 40°N. Vertical pink bars in (a) denote strong periods (SP). (b) Normalized power spectrum density of (a) (kyr). Vertical pink bars in (b) denote the 23-kyr, 41-kyr, and 100-kyr cycles.

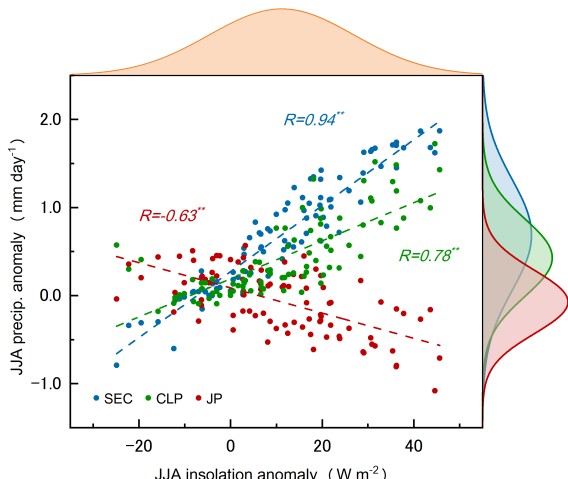

**Figure 3: Correlation between the simulated JJA insolation (W m$^{-2}$) and precipitation (mm day$^{-1}$) anomalies of three EASM regions (SEC, CLP, and JP).** Insolation was averaged over 20° N–40° N. Figures in the diagram show correlation coefficients. The shaded curves in the top and right show the normal distribution obtained from the mean and standard deviation of each data.

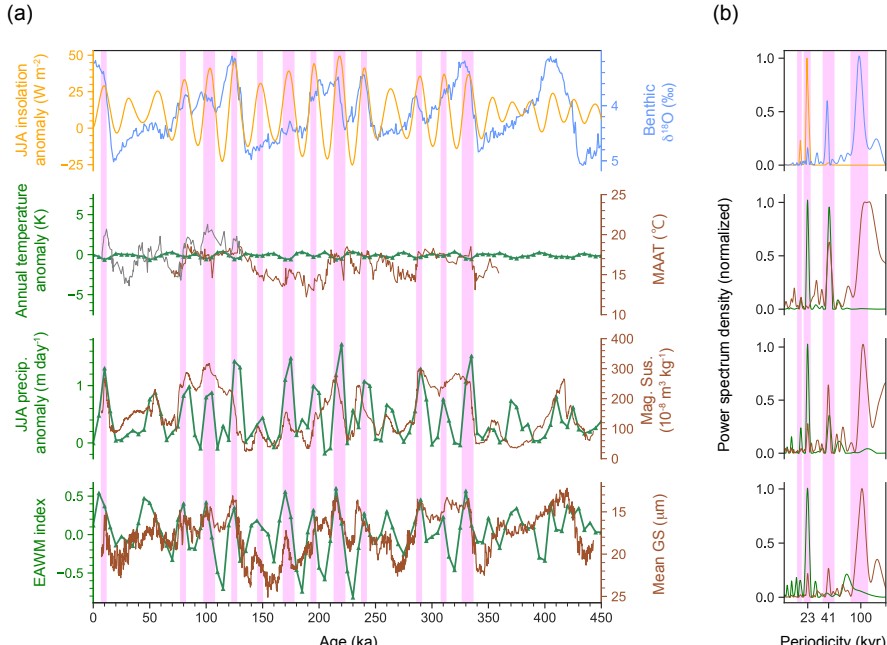

**Figure 4: Model-proxy comparison in CLP.** (a) Temporal variation in JJA insolation anomaly (orange, W m$^{-2}$) as depicted in Fig. 2, benthic $\delta^{18}$O stack (light blue, ‰, Lisiecki and Raymo, 2005), simulated annual temperature (K), and JJA precipitation (mm day$^{-1}$) anomaly in CLP (green), East Asian Winter Monsoon index based on Wang and Cheng (2014). Insolation was averaged over 20° N–40° N. Brown lines indicate proxy records: Mean annual air temperature reconstructed from brGDGT (°C), magnetic susceptibility (10$^{-8}$ m$^3$ kg$^{-1}$), and mean grain size (mm) (Thomas et al. 2016). Vertical pink bars in (a) denote strong periods (SP). (b) Normalized power spectrum density of (a) (kyr). Vertical pink bars in (b) denote the 23-kyr, 41-kyr, and 100-kyr cycles.

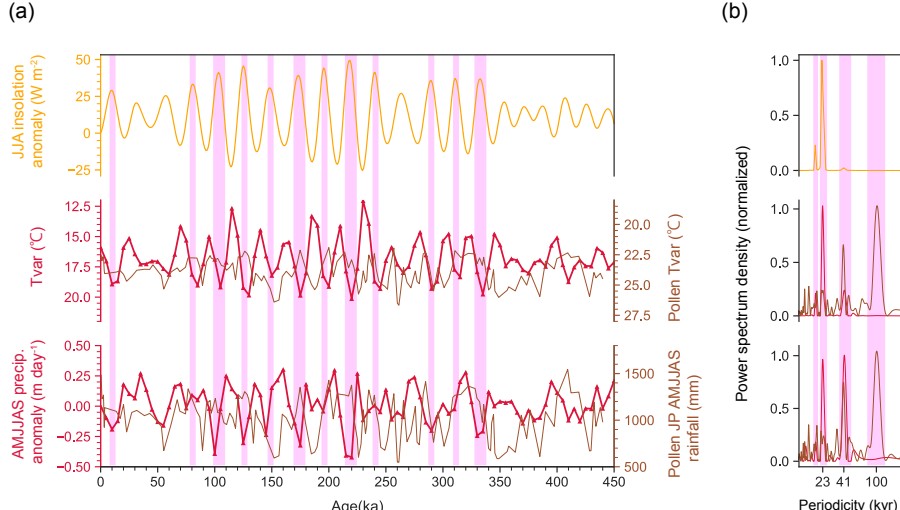

**Figure 5: Model-proxy comparison in Japan (JP).** (a) Temporal variation in simulated JJA insolation anomaly (orange, W m$^{-2}$) as depicted in Fig. 2, annual temperature range Tvar (K), and AMJJAS precipitation (mm day$^{-1}$) anomaly in CLP (red). Insolation was averaged over 20° N–40° N. Brown lines indicate proxy records: Tvar (°C) and AMJJAS rainfall (mm). Vertical pink bars in (a) denote strong periods (SP). (b) Normalized power spectrum density of (a) (kyr). Vertical pink bars in (b) denote the 23-kyr, 41-kyr, and 100-kyr cycles.

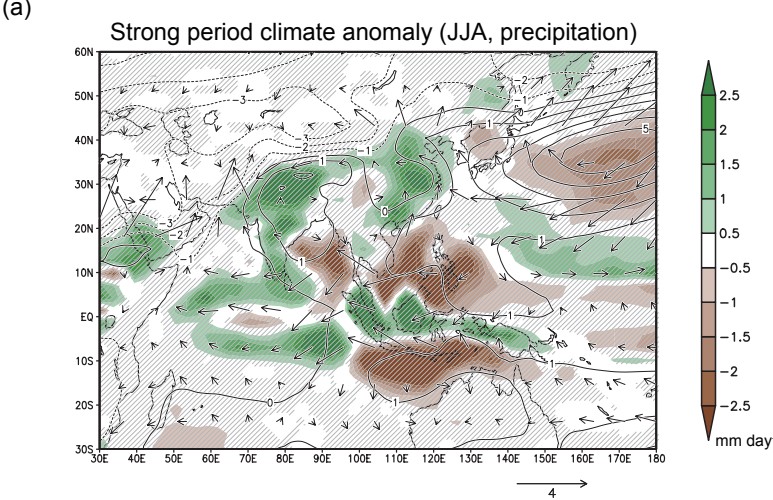

(a) Strong period climate anomaly (JJA, precipitation)

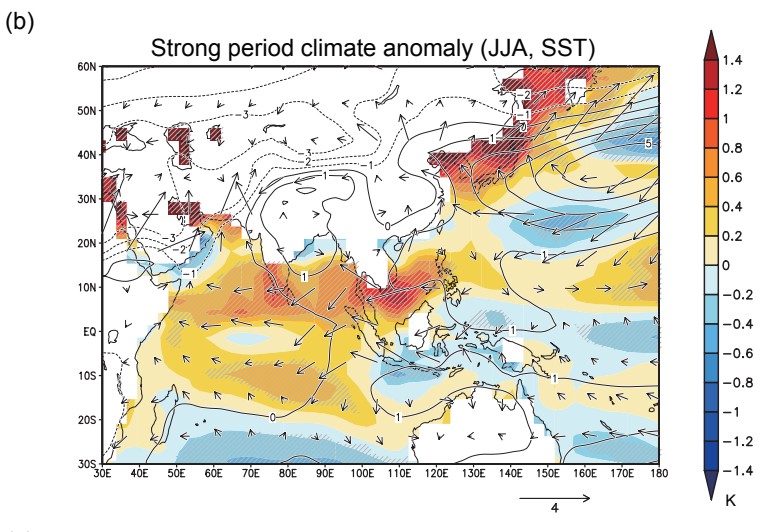

(b) Strong period climate anomaly (JJA, SST)

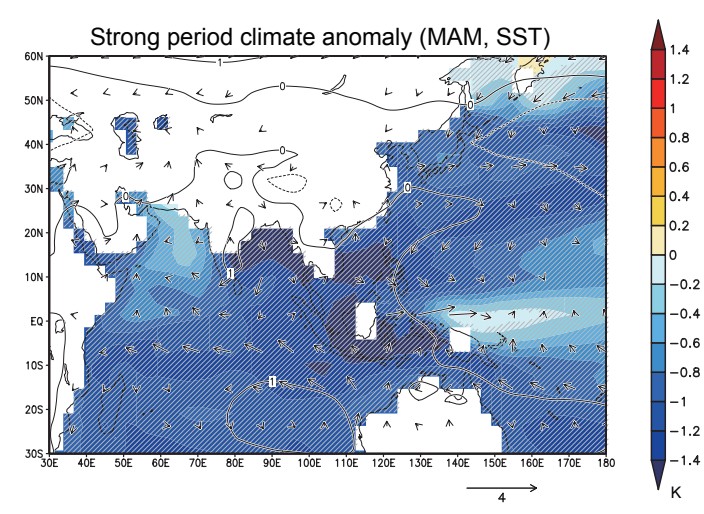

(c) Strong period climate anomaly (MAM, SST)

**Figure 6: (a)** Spatial distribution of the JJA precipitation (shading, mm day$^{-1}$), sea level pressure (contour, hPa), and surface wind (vector, m s$^{-1}$) anomalies in strong periods. Solid and dashed lines denote positive and negative, respectively. Surface wind anomaly under 0.1 m s$^{-1}$ is masked. Hatched area denotes regions exceeding a 95 % confidence level using Student's t-test. **(b)** Spatial distribution of the JJA sea surface temperature (shading, K), sea level pressure (contour, hPa), and surface wind (vector, m s$^{-1}$) anomalies in strong periods. Solid and dashed lines denote positive and negative, respectively. Surface wind anomaly under 0.1 m s$^{-1}$ is masked. Hatched area denotes regions exceeding a 95 % confidence level using Student's t-test. **(c)** Spatial distribution of spring (MAM) sea surface temperature (shading, K), sea level pressure (contour, hPa), and surface wind (vector, m s$^{-1}$) anomalies in strong periods. Solid and dashed lines denote positive and negative, respectively. Surface wind anomaly under 0.1 m s$^{-1}$ is masked. Hatched area denotes regions exceeding a 95 % confidence level using Student's t-test. Note that the colour bar in this diagram differs from Fig. 5.

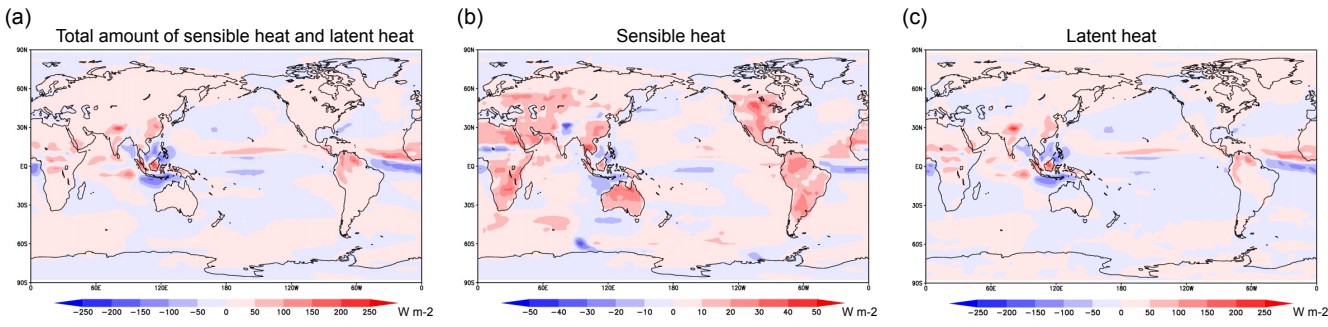

**Figure 7: Anomaly of JJA vertically integrated diabatic heating.** (a) The total amount of sensible heat and latent heat, (b) sensible heat, and (c) latent heat. All units are W m⁻².