# Peer review of "Contrasting responses of summer precipitation to orbital forcing in Japan and China over the past 450 kyr"

_EGUsphere, 2024_

## Referee Comment (RC1)

**General comments:**

The work titled "The Effects of Orbital Forcing on the East Asian Summer Monsoon for the Past 450 kyr" is intriguing. The authors have adeptly reviewed previous research progress and provided a detailed and clear explanation of the research methodology in this paper. However, the section discussing their own research findings appears to be rather weak, lacking a comprehensive showcase of the work's novel discoveries. Substantial revisions are necessary to better highlight the new contributions of the study.

**Specific comments to the authors**

The term "calculate and calculation" in the given sentence (Abstract) is not accurate; it should be replaced with "simulated" or "simulation." Please check similar issue thorough the manuscript.

The initial segment of the abstract is well-structured; however, the latter part, starting from "The calculated change in summer precipitation is dominated by a 20-kyr precession cycle over China, highly consistent with cave d18O records in southeast China," becomes overly generalized. The author delves into various aspects, addressing the periodicity of simulated East Asian Summer Monsoon (EASM) precipitation in connection with forcing cycles. Subsequently, a correlation analysis is presented to establish the relationship between EASM precipitation intensity and solar radiation forcing. This deviates somewhat from the conventional approach of enhancing mechanistic understanding through numerical simulations. Therefore, in this section, I recommend that the author enrich the paper by incorporating more explanations related to climate dynamics.

The Introduction section lacks a recent review of the advancements in the comparison of data and models in East Asian paleomonsoonal dynamics.

Sun, Y., H. Wu, G. Ramstein, B. Liu, Y. Zhao, L. Z. X. Li, X. Y. Yuan, W. C. Zhang, L. J. Li, L. W. Zou, T. J. Zhou. Revisiting the Physical Mechanisms of East Asian Summer Monsoon Precipitation Changes During the Mid-Holocene: A Data–model Comparison. Climate Dynamics 60, 1009–1022 (2023). https://doi.org/10.1007/s00382-022-06359-1.

Sun, Y., H. Wu, M. Kageyama, G. Ramstein, L. Z. X. Li, N. Tan, Y. T. Lin, B. Liu, W. P. Zheng, W. C. Zhang, L. W. Zou, T. J. Zhou. 2021. The contrasting effects of thermodynamic and dynamic processes on East Asian summer monsoon precipitation during the Last Glacial Maximum: a data-model comparison. Climate Dynamics. 56, 1303–1316.

Sun, Y., G. Ramstein, L. Z. X. Li, C. Contoux, N. Tan, T. J. Zhou. 2018. Quantifying East Asian summer monsoon dynamics in the ECP4.5 scenario with reference to

the mid-Piacenzian warm period. Geophysical Research Letters, 45: 12,523–12,533.

I could not agree with the authors statements "Section 4 discusses the possible climate systems that drive EASM variability". As we knew, orbital forcing via solar radiation changes can be attributed fundamental driver of climate changes, here the authors may discuss the possible climate systems associated with EASM variability.

L86: "due to orbital forcing" needs to put behind the insolation changes

L131-135 should move to the method section somewhere.

Title in section 3.1 is confusing, if I understand well the authors want to express "simulated······"?

I have additional comments on the organization of the results section. In fact, it is not necessary to divide Section 3 into two subsections. The authors intend to focus on one specific task in this section: the model-data comparison of East Asian Summer Monsoon (EASM) precipitation evolution for the last 450,000 years. The current version contains numerous citations, making it challenging for the reader and reviewer to discern the extent of the authors' new findings. Consolidating the section into a single subsection may help clarify the presentation and emphasize the novel contributions of the authors. Please rephase these sections.

Figure.4-5-6 can be merged into one new Figure.

L427: please use SEC instead South East China, as the abbreviation has already appeared.

---

## Author Comment (AC1)

**Responses to Reviewers' comments**

We are grateful for the careful review and valuable feedback provided by the referees. Below, we provide responses to each of the reviewers' comments and indicate our plans to revise the manuscript. Comments from the referees are marked in *italics*, and our detailed responses to each comment are in blue font. We hope that the revision addresses the referees' concerns.

**Reviewer 1**

**Comment 1:** *The work titled "The Effects of Orbital Forcing on the East Asian Summer Monsoon for the Past 450 kyr" is intriguing. The authors have adeptly reviewed previous research progress and provided a detailed and clear explanation of the research methodology in this paper. However, the section discussing their own research findings appears to be rather weak, lacking a comprehensive showcase of the work's novel discoveries. Substantial revisions are necessary to better highlight the new contributions of the study.*

**Response to comment 1:** We appreciate the reviewer's constructive feedback. We believe the novelty of our study lies in demonstrating the varying impact of solar radiation variability on the East Asian monsoon across different regions. Particularly noteworthy is the discovery that summer precipitation patterns in Japan and China exhibit distinct responses, each governed by different atmospheric circulation mechanisms—the reinforced North Pacific High and sub-high, respectively. Based on your suggestions in Comment 1 and Comment 9, we will revise sections 3 and 4 to underscore the novelty of this study. Additionally, in response to Reviewer 2's suggestion, we will include simulation results beyond summer precipitation and incorporate additional proxy data for comparison in the revised paper. We will include simulated annual precipitation and summer temperature, which will be compared with simulated summer precipitation and proxies (speleothem $\delta^{18}O$, $^{10}Be$ and pollen records). We will also extend the model-proxy comparison in CLP and JP, by incorporating several other proxies that offer access to variabilities of annual mean temperature (CLP), winter monsoon (CLP) and annual temperature range (JP). These changes will alter Fig.2; additionally, we will add new figures, Fig.4 and Fig.5, as well as a new discussion on model validation and proxy-data comparison. These new figures are at the end of this manuscript.

**Comment 2:** *The term "calculate and calculation" in the given sentence (Abstract) is not accurate; it should be replaced with "simulated" or "simulation." Please check similar issue thorough the manuscript.*

**Response to comment 2:** These will be corrected as suggested.

**Comment 3:** *The initial segment of the abstract is well-structured; however, the latter part, starting from "The calculated change in summer precipitation is dominated by a 20-kyr precession cycle over China, highly consistent with cave d18O records in southeast China," becomes overly generalized. The author delves into various aspects, addressing the periodicity of simulated East Asian Summer Monsoon (EASM) precipitation in connection with forcing cycles. Subsequently, a correlation analysis is presented to establish the relationship between EASM precipitation intensity and solar radiation forcing. This deviates somewhat from the conventional approach of enhancing mechanistic understanding through numerical simulations. Therefore, in this section, I recommend that the author enrich the paper by incorporating more explanations related to climate dynamics.*

**Response to comment 3:** We thank the referee's constructive comment. Reviewer 2 also suggested rewriting the abstract. Considering the feedback from the two reviewers, we will rework the latter portion of the abstract. Specifically, we will revise the structure and wording to highlight the simulation outcomes: the influence of solar radiation forcing on summer precipitation across each East Asian monsoon region (SEC, CLP, and JP). The description of correlation analyses will be deleted from the abstract. Subsequently, we will delve into the underlying causes of simulated EASM variability from a climate dynamics perspective.

**Comment 4:** *The Introduction section lacks a recent review of the advancements in the comparison of data and models in East Asian paleomonsoonal dynamics.*

*Sun, Y., H. Wu, G. Ramstein, B. Liu, Y. Zhao, L. Z. X. Li, X. Y. Yuan, W. C. Zhang, L. J. Li, L. W. Zou, T. J. Zhou. Revisiting the Physical Mechanisms of East Asian Summer Monsoon Precipitation Changes During the Mid-Holocene: A Data–model Comparison. Climate Dynamics 60, 1009–1022 (2023). https://doi.org/10.1007/s00382-022-06359-1.*

*Sun, Y., H. Wu, M. Kageyama, G. Ramstein, L. Z. X. Li, N. Tan, Y. T. Lin, B. Liu, W. P. Zheng, W. C. Zhang, L. W. Zou, T. J. Zhou. 2021. The contrasting effects of thermodynamic and dynamic processes on East Asian summer monsoon precipitation during the Last Glacial Maximum: a data-model comparison. Climate Dynamics. 56, 1303–1316.*

*Sun, Y., G. Ramstein, L. Z. X. Li, C. Contoux, N. Tan, T. J. Zhou. 2018. Quantifying East Asian summer monsoon dynamics in the ECP4.5 scenario with reference to the mid-Piacenzian warm period. Geophysical Research Letters, 45: 12,523–12,533.*

**Response to comment 4:** We thank the referee for providing information about these papers. We will incorporate them into the introduction.

**Comment 5:** *I could not agree with the authors statements "Section 4 discusses the possible climate systems that drive EASM variability". As we knew, orbital forcing via solar radiation changes can be attributed fundamental driver of climate changes, here the authors may discuss the possible climate systems associated with EASM variability.*

**Response to comment 5:** We will revise the phrasing accordingly.

**Comment 6:** *L86: "due to orbital forcing" needs to put behind the insolation changes*

**Response to comment 6:** Will be corrected as suggested.

**Comment 7:** *L131-135 should move to the method section somewhere.*

**Response to comment 7:** Will be corrected as suggested.

**Comment 8:** Title in section 3.1 is confusing, if I understand well the authors want to express "simulated……"?

**Response to comment 8:** Will be corrected.

**Comment 9:** *I have additional comments on the organization of the results section. In fact, it is not necessary to divide Section 3 into two subsections. The authors intend to focus on one specific task in this section: the model-data comparison of East Asian Summer Monsoon (EASM) precipitation evolution for the last 450,000 years. The current version contains numerous citations, making it challenging for the reader and reviewer to discern the extent of the authors' new findings. Consolidating the section into a single subsection may help clarify the presentation and emphasize the novel contributions of the authors. Please rephase these sections.*

**Response to comment 9:** As suggested, Section 3 will be combined into one section instead of being divided into subsections and will be modified to emphasize the novelty of this study.

**Comment 10:** *Figure.4-5-6 can be merged into one new Figure.*

**Response to comment 10:** Will be modified as suggested.

**Comment 11:** *L427: please use SEC instead South East China, as the abbreviation has already appeared.*

**Response to comment 11:** Will be corrected.

**Reviewer 2**

**Comment 1:** *This paper presented new results about how orbital forcing influence the East Asian Summer Monsoon by a group of new time-slice simulations. The authors conducted an extensive review of previous research. But more discussion should be added regarding their own results. (1) They only presented summer precipitation changes. But for East Asian summer monsoon, annual precipitation and summer temperature could also be presented and compared with proxy records; (2) They only show three proxy records. More model-proxy comparison should be added.*

**Response to comment 1:** We appreciate this feedback. Recognizing the significance of enriching the context of our simulation outcomes, we will incorporate additional data and discussion. In response to the concerns raised, we will make the following modifications in the revised manuscript:

(1) We will include the simulation results of annual precipitation and summer temperature variability in the three regions under investigation in our study (SEC, CLP, and JP). The simulation results will be compared with simulated summer precipitation and summer/annual precipitation proxies.

(2) Additionally, we will expand the dataset by incorporating several other proxies in CLP and JP that offer access to variabilities of annual mean temperature (CLP), winter monsoon (CLP), and annual temperature range (JP).

In response to Reviewer 1's suggestion, we will also modify the structure of section 3 and revise the sentences in section 3 and 4 to underscore the novelty of this study. These changes will alter Fig.2, additionally, we will add new figures, Fig.4 and Fig.5. We will also add a new discussion on model validation and proxy-data comparison based on new Fig. 2, Fig.4 and Fig.5. These new figures are at the end of this response file.

**Comment 2:** *The abstract is a bit confusing and should be rewritten.*

**Response to comment 2:** The abstract will be revised accordingly. Reviewer 1 also recommended revising the abstract. Taking into consideration the feedback from the reviewers, we will rewrite the latter portion of the abstract to enrich the explanation from the standpoint of climate dynamics and accentuate the novel facets of this study.

**Comment 3:** *Line 18 'Calculated' should be 'Simulated'.   Similar expressions throughout the text need to be modified.*

**Response to comment 3:** These will be corrected as suggested.

**Comment 4:** *Line 41 23 kyr periodicity should be 23-kyr periodicity. Similar expressions throughout the text need to be modified, e,g. line 43.*

**Response to comment 4:** Will be corrected as suggested.

**Comment 5:** *Line 34 Reference should be (An et al., 2015)*

**Response to comment 5:** Will be corrected as suggested.

**Comment 6:** *Line 46 Logic question. How does 'EASM varies in phase with orbital cycles' suggest 'the EASM is affected by changes in ice volume'? The author seems to confuse '100-kyr cycle' and 'eccentricity cycle'.*

**Response to comment 6:** In this sentence we assumed that the eccentricity cycle is the underlying cause of the 100-kyr glacial-interglacial cycle, as written in Line 42−45. In response to the comment raised, we will rewrite the sentence in the revised manuscript, to avoid confusion between '100-kyr cycle' that exist in ice-volume, which can be a direct forcing factor for EASM, and the orbital 'eccentricity cycle':

> "Geological records indicate that the EASM is affected by changes in insolation or ice volume, or both (e.g., Cheng et al., 2016; Clemens et al., 2018; G. Liu et al., 2020; Sun et al., 2006, 2015, 2019) "

In the revised sentence we erased "the EASM varies in phase with orbital cycles," from the original sentence. As we address the periodicity of each geological record and corresponding possible climatic forcings in the following sentences, removing the phrase will not change our overall conclusions.

**Comment 7:** *Line 52 Rewrite the sentence.*

**Response to comment 7:** Thank you for the feedback. The sentence will be rewritten.

**Comment 8:** *Line 61 Rewrite the sentence.*

**Response to comment 8:** Thank you for the feedback. The sentence will be rewritten.

**Comment 9:** *Line 20 I do not agree that the results can come to the conclusion ... the importance of other forcing (e.g. ice sheet). Besides external forcings, the internal feedback should also be considered.*

**Response to comment 9:** We thank the referee for bringing this to our attention. It was not our intention to disregard the potential effects of internal feedback mechanisms. We acknowledge the significance of considering both external forcing and internal feedback in interpreting geological records. This sentence will be revised.

**Comment 10:** *Line 97 This model comprises an atmospheric general circulation model (AGCM) and a global ocean general circulation model (OGCM). This sentence could be deleted because everyone knows that a coupled GCM consists of AGCM and OGCM.*

**Reply to comment 10:** Will be corrected as suggested.

**Comment 11:** *More introduction on the model should be added, e,g, if there is ice-sheet model.*

**Response to comment 11:** We will provide additional elaboration on the land model and the sea ice simulation, specifically addressing the treatment of the ice sheet.

**Comment 12:** *Line 171 Why is the 100-kyr band from eccentricity? Why not ice volume?*

**Response to comment 12:** As clarified in our response to comment 6, this presupposes that the eccentricity cycle is responsible for the 100-kyr glacial-interglacial cycle. We will revise the sentence to prevent confusion between eccentricity cycle and ice-volume variability.

**Figures**

[Figure]

**New Fig. 2** (a) Temporal variation in simulated JJA insolation anomaly (orange, W m-2), precipitation anomaly in SEC (blue), CLP (green), and JP (red). Insolation was averaged over 20° N–40° N. The precipitation anomaly unit is mm day[-1.] Brown lines indicate proxy records: Speleothem $\delta^{18}O$ over SEC (‰, Cheng et al., 2016), [10]Be-based annual rainfall over CLP (mm, Beck et al., 2018), and pollen-based AMJJAS rainfall variability in Lake Biwa in JP (K, Nakagawa et al., 2008). Grey lines indicate simulated annual temperature anomaly (K) in each region. Vertical pink bars in (a) denote strong periods (SP). (b) Normalized power spectrum density of (a) (kyr). Vertical pink bars in (b) denote the main orbital cycles (precession, obliquity, and eccentricity).

[Figure]

**New Fig. 4** Model-proxy comparison in CLP. (a) Temporal variation in simulated JJA insolation anomaly (orange, W m$^{-2}$), benthic δ$^{18}$O stack (light blue, ‰, Lisiecki and Raymo, 2005), annual temperature (K) and JJA precipitation (mm day$^{-1}$) anomaly in CLP (green), East Asian Winter Monsoon index based on Wang and Cheng (2014). Insolation was averaged over 20° N–40° N. Brown lines indicate proxy records: Mean Annual Air Temperature reconstructed from brGDGT (°C), Magnetic Susceptibility (10$^{-8}$ m$^3$ kg$^{-1}$), and Mean Grain Size (μm) (Thomas et al. 2016). Vertical pink bars in (a) denote strong periods (SP) as in Fig. 2. (b) Normalized power spectrum density of (a) (kyr). Vertical pink bars in (b) denote the main orbital cycles (precession, obliquity, and eccentricity).

[Figure]

**New Fig. 5** Model-proxy comparison in Japan (JP). (a) Temporal variation in simulated JJA insolation anomaly (orange, W m$^{-2}$), annual temperature range Tvar (K) and AMJJAS precipitation (mm day$^{-1}$) anomaly in CLP (red). Insolation was averaged over 20° N–40° N. Brown lines indicate proxy records: Tvar (°C) and AMJJAS rainfall (mm). Vertical pink bars in (a) denote strong periods (SP) as in Fig. 2 and Fig. 4. (b) Normalized power spectrum density of (a) (kyr). Vertical pink bars in (b) denote the main orbital cycles (precession, obliquity, and eccentricity).

---

## Author Response (AR1)

**Responses to Reviewers' comments**

We are grateful for the careful review and valuable feedback provided by the referees. Below, we have provided our point-by-point responses to each of the reviewers' comments and detail the associated revisions made to the manuscript. Comments from the referees are marked in *italics*, and our responses are in blue font. Revisions made to the manuscript are highlighted in yellow for your convenience. We hope that the revision addresses the referees' concerns.

**Reviewer 1**

**Comment 1:** *The work titled "The Effects of Orbital Forcing on the East Asian Summer Monsoon for the Past 450 kyr" is intriguing. The authors have adeptly reviewed previous research progress and provided a detailed and clear explanation of the research methodology in this paper. However, the section discussing their own research findings appears to be rather weak, lacking a comprehensive showcase of the work's novel discoveries. Substantial revisions are necessary to better highlight the new contributions of the study.*

**Response to comment 1:** We appreciate the reviewer's constructive feedback. We believe the novelty of our study lies in demonstrating the varying impact of solar radiation variability on the East Asian monsoon across different regions. Particularly noteworthy is the discovery that summer precipitation patterns in Japan and China exhibit distinct responses, each governed by different atmospheric circulation mechanisms—the reinforced North Pacific High and sub-high, respectively. However, based on your suggestions in Comment 1 and Comment 9, we revised sections 3 and 4 to underscore the novelty of this study. Additionally, in response to Reviewer 2's suggestion, we included simulation results beyond summer precipitation and incorporated additional proxy data for comparison. We have included simulated annual precipitation and summer temperature, which was compared with insolation, simulated summer precipitation, and proxies (speleothem $\delta^{18}$O, $^{10}$Be, and pollen records). Furthermore, we have extended the model-proxy comparison in CLP and JP by incorporating several other proxies that offer access to variabilities in annual mean temperature (CLP), winter monsoon (CLP), and annual temperature range (JP). These revisions have altered the data presented in Fig. 2; additionally, we have added Fig. 4 and Fig. 5, as well as a supplemented discussion on model validation and proxy-data comparison.

**Comment 2:** *The term "calculate and calculation" in the given sentence (Abstract) is not accurate; it should be replaced with "simulated" or "simulation." Please check similar issue thorough the manuscript.*

**Response to comment 2:** The text was revised per your suggestion.

**Comment 3:** *The initial segment of the abstract is well-structured; however, the latter part, starting from "The calculated change in summer precipitation is dominated by a 20-kyr precession cycle over China, highly consistent with cave d18O records in southeast China," becomes overly generalized. The author delves into various aspects, addressing the periodicity of simulated East Asian Summer Monsoon (EASM) precipitation in connection with forcing cycles. Subsequently, a correlation analysis is presented to establish the relationship between EASM precipitation intensity and solar radiation forcing. This deviates somewhat from the conventional approach of enhancing mechanistic understanding through numerical simulations. Therefore, in this section, I recommend that the author enrich the paper by incorporating more explanations related to climate dynamics.*

**Response to comment 3:** We thank the referee for the constructive comment. Reviewer 2 also suggested rewriting the abstract. Accordingly, we have reworked the latter portion of the abstract. Specifically, we revised the structure and wording to highlight the simulation outcomes: the influence of solar radiation on summer precipitation across each East Asian monsoon region (SEC, CLP, and JP). Additionally, the description of correlation analyses was removed from the abstract. Subsequently, we delved into the underlying causes of simulated EASM variability from a climate dynamics perspective.

**Comment 4:** *The Introduction section lacks a recent review of the advancements in the comparison of data and models in East Asian paleomonsoonal dynamics.*

*Sun, Y., H. Wu, G. Ramstein, B. Liu, Y. Zhao, L. Z. X. Li, X. Y. Yuan, W. C. Zhang, L. J. Li, L. W. Zou, T. J. Zhou. Revisiting the Physical Mechanisms of East Asian Summer Monsoon Precipitation Changes During the Mid-Holocene: A Data–model Comparison. Climate Dynamics 60, 1009–1022 (2023). https://doi.org/10.1007/s00382-022-06359-1.*

*Sun, Y., H. Wu, M. Kageyama, G. Ramstein, L. Z. X. Li, N. Tan, Y. T. Lin, B. Liu, W. P. Zheng, W. C. Zhang, L. W. Zou, T. J. Zhou. 2021. The contrasting effects of thermodynamic and dynamic processes on East Asian summer monsoon precipitation during the Last Glacial Maximum: a data-model comparison. Climate Dynamics. 56, 1303–1316.*

*Sun, Y., G. Ramstein, L. Z. X. Li, C. Contoux, N. Tan, T. J. Zhou. 2018. Quantifying East Asian summer monsoon dynamics in the ECP4.5 scenario with reference to the mid-Piacenzian warm period. Geophysical Research Letters, 45: 12,523–12,533.*

**Response to comment 4:** Thank you for providing these additional relevant articles, which have been incorporated into the introduction.

**Comment 5:** *I could not agree with the authors statements "Section 4 discusses the possible climate systems that drive EASM variability". As we knew, orbital forcing via solar radiation changes can be attributed fundamental driver of climate changes, here the authors may discuss the possible climate systems associated with EASM variability.*

**Response to comment 5:** The phrasing has been revised accordingly.

**Comment 6:** *L86: "due to orbital forcing" needs to put behind the insolation changes*

**Response to comment 6:** The text has been revised per your suggestion.

**Comment 7:** *L131-135 should move to the method section somewhere.*

**Response to comment 7:** The text has been revised per your suggestion.

**Comment 8:** Title in section 3.1 is confusing, if I understand well the authors want to express "simulated……"?

**Response to comment 8:** The text has been revised per your suggestion.

**Comment 9:** *I have additional comments on the organization of the results section. In fact, it is not necessary to divide Section 3 into two subsections. The authors intend to focus on one specific task in this section: the model-data comparison of East Asian Summer Monsoon (EASM) precipitation evolution for the last 450,000 years. The current version contains numerous citations, making it challenging for the reader and reviewer to discern the extent of the authors' new findings. Consolidating the section into a single subsection may help clarify the presentation and emphasize the novel contributions of the authors. Please rephase these sections.*

**Response to comment 9:** As suggested, Section 3 has been presented as a single section rather than being divided into subsections and was modified to emphasize the study's novelty.

**Comment 10:** *Figure.4-5-6 can be merged into one new Figure.*

**Response to comment 10:** The text has been revised per your suggestion.

**Comment 11:** *L427: please use SEC instead South East China, as the abbreviation has already appeared.*

**Response to comment 11:** The text has been revised per your suggestion.

**Reviewer 2**

**Comment 1:** *This paper presented new results about how orbital forcing influence the East Asian Summer Monsoon by a group of new time-slice simulations. The authors conducted an extensive review of previous research. But more discussion should be added regarding their own results. (1) They only presented summer precipitation changes. But for East Asian summer monsoon, annual precipitation and summer temperature could also be presented and compared with proxy records; (2) They only show three proxy records. More model-proxy comparison should be added.*

**Response to comment 1:** We appreciate this feedback. Recognizing the significance of enriching the context of our simulation outcomes, we have incorporated additional data and discussion. In response to the concerns raised, we made the following modifications to the revised manuscript:

(1) Annual precipitation and summer temperature variability simulation results have been included for the three regions under investigation (SEC, CLP, and JP). The simulation results were compared with simulated summer precipitation and summer/annual precipitation proxies.

(2) The dataset was expanded by incorporating several other proxies in CLP and JP that offer access to variabilities of annual mean temperature (CLP), winter monsoon (CLP), and annual temperature range (JP).

In response to Reviewer 1's suggestion, we also modified the structure of Section 3 and revised the associated text in Sections 3 and 4 to underscore the study's novelty. These revisions have altered the data presented in Fig. 2; additionally, we have added Fig. 4 and Fig. 5. We have also added a new discussion on model validation and proxy-data comparison based on the revised Fig. 2, Fig. 4, and Fig. 5.

**Comment 2:** *The abstract is a bit confusing and should be rewritten.*

**Response to comment 2:** The abstract was revised accordingly. Taking into consideration the feedback from both reviewers, we have rewritten the latter portion of the abstract to enrich the explanation from the standpoint of climate dynamics and accentuate the novel facets of this study.

**Comment 3:** *Line 18 'Calculated' should be 'Simulated'. Similar expressions throughout the text need to be modified.*

**Response to comment 3:** The text has been revised per your suggestion.

**Comment 4:** *Line 41 23 kyr periodicity should be 23-kyr periodicity. Similar expressions throughout the text need to be modified, e,g. line 43.*

**Response to comment 4:** The text has been revised per your suggestion.

**Comment 5:** *Line 34 Reference should be (An et al., 2015)*

**Response to comment 5:** The text has been revised per your suggestion.

**Comment 6:** *Line 46 Logic question. How does 'EASM varies in phase with orbital cycles' suggest 'the EASM is affected by changes in ice volume'? The author seems to confuse '100-kyr cycle' and 'eccentricity cycle'.*

**Response to comment 6:** In this sentence, we assumed that the eccentricity cycle is the underlying cause of the 100-kyr glacial–interglacial cycle (Lines 42−44 in the reviesed manuscript). In response to the comment raised, the text has been revised as follows to avoid confusion between the '100-kyr cycle' that exists in ice volume, which can be a direct forcing factor for EASM, and the orbital 'eccentricity cycle':

> "Geological records indicate that the EASM is affected by changes in insolation, ice volume, or both (e.g., Cheng et al., 2016; Clemens et al., 2018; G. Liu et al., 2020; Sun et al., 2006, 2015, 2019) " (Lines 46−47 in the reviesed manuscript)

The sentence "the EASM varies in phase with orbital cycles" was removed. As we address the periodicity of each geological record and corresponding possible climatic forcing in the subsequent text, removing this phrase will not alter the overall conclusions.

**Comment 7:** *Line 52 Rewrite the sentence.*

**Response to comment 7:** Thank you for the feedback. The sentence was rewritten accordingly.

**Comment 8:** *Line 61 Rewrite the sentence.*

**Response to comment 8:** Thank you for the feedback. The text was revised accordingly.

**Comment 9:** *Line 20 I do not agree that the results can come to the conclusion ... the importance of other forcing (e.g. ice sheet). Besides external forcings, the internal feedback should also be considered.*

**Response to comment 9:** We thank the referee for bringing this to our attention. It was not our intention to disregard the potential effects of internal feedback mechanisms. We acknowledge the significance of considering external forcing and internal feedback when interpreting geological records. The manuscript has been revised to reflect this.

**Comment 10:** *Line 97 This model comprises an atmospheric general circulation model (AGCM) and a global ocean general circulation model (OGCM). This sentence could be deleted because everyone knows that a coupled GCM consists of AGCM and OGCM.*

**Reply to comment 10:** The text has been revised per your suggestion.

**Comment 11:** *More introduction on the model should be added, e,g, if there is ice-sheet model.*

**Response to comment 11:** We have supplemented our description of the land model and sea ice simulation, specifically addressing the treatment of the ice sheet:

> "The land component, incorporating vegetation effects, relies on the Simple Biosphere (SiB) model (Sellers et al., 1986; Sato et al., 1989). In this model, the ice sheet is treated analogously to vegetation. However, the dynamics of vegetation and the ice sheet are not integrated; adjustments in ice volume can be accommodated by modifying land vegetation and altitude, specified as boundary conditions. Sea ice compactness and thickness are forecasted utilising thermodynamics, horizontal advection, and diffusion principles." (Lines 101−105 in the reviesed manuscript)

**Comment 12:** *Line 171 Why is the 100-kyr band from eccentricity? Why not ice volume?*

**Response to comment 12:** As clarified in our response to comment 6, this presupposes that the eccentricity cycle is responsible for the 100-kyr glacial–interglacial cycle. We have revised the text to avoid confusion between the eccentricity cycle and ice volume variability. (Line 189 in the reviesed manuscript)

---

## Author Response (AR2)

**Responses to Editor's comments**

We are grateful for the careful review provided by the Editor. Below, we have provided our point-by-point responses to each of the comments and have detailed the associated revisions made to the manuscript. Comments from the Editor are marked in *italics*, and our responses are indicated by blue-coloured text. Revisions made to the manuscript are highlighted in yellow for your convenience. We hope that the revisions we have made to the manuscript address the Editor's concerns.

**Comment 1:** *This title seems to me too general (and also because there exist many similar studies), and does not reflect the novelty of this paper which is mainly on the different response to insolation between China and Japan. I would suggest to change a title to reflect more the novelty of this study.*

**Response to comment 1:** We appreciate the Editor's constructive feedback. We have changed the title of the manuscript as follows:

**"Contrasting responses of summer precipitation to orbital forcing in Japan and China over the past 450 kyr".**

**Comment 2:** *are they eccentricity and obliquity cycles or ~100-kyr and ~40-kyr cycles?*

**Response to comment 2:** We have corrected the text to 100-kyr and 40-kyr cycles. (Line 50 in the revised manuscript)

**Comment 3:** *It is not clear for me. Could you give more information on the vegetation model and the ice sheet component?*

**Response to comment 3:** In this model, ice sheets are treated as one of the vegetation types which is reflected in the albedo in the simulation. Vegetation is fixed as a boundary condition in each simulation. The text has been modified to clarify these points. (Lines 102−103 in the revised manuscript)

**Comment 4:** *this is a fact. This sentence might need to be rephrased.*

**Response to comment 4:** The phrasing has been revised to state this as fact. (Lines 163−164 in the revised manuscript)

**Comment 5:** *56 kyr is one of the periodicities of precession although its amplitude is much weaker than the 19 and 23 kyr (See Berger, 1978, Table 2)*

**Response to comment 5:** Thank you for pointing this out. We have added your remarks to the revised text. (Lines 167−168 in the revised manuscript)

**Comment 6:** *This expression is not understandable for me.*

**Response to comment 6:** The text has been revised for clarity. (Line 215 in the revised manuscript)

**Comment 7:** *precipitation change on land is not shown on Fig.6a*

**Comment 8:** *Fig.6a shows decreasing precipitaiton over the Indian Ocean*

**Response to comment 7 and 8:** Thank you for pointing this out. During the last revision, We mistakenly plotted the spring (MAM) SST deviation (same figure as Fig. 6c) in Fig. 6a; we have thus replaced Fig. 6a with the correct figure. The corrected Fig. 6a now shows precipitation anomalies over land, as well as the increase in precipitation in the Indian Ocean.

**Comment 9:** *is it shown in Fig.6?*

**Response to comment 9:** In the corrected Fig. 6a and Fig. 6b, anomalous easterly winds are observed from the Indian Ocean to the western North Pacific around Philippines. We have revised the text to clarify these points. (Line 236−237 in the revised manuscript)

**Comment 10:** *why spring?*

**Response to comment 10:** We checked the climate anomaly in spring because negative SST anomalies in the tropical northwestern Pacific during spring can lead to the formation and maintenance of sub-highs through wind evaporation SST (WES) feedback. These mechanisms are referred to as the Indo-western Pacific Ocean Capacitor (IPOC) mode. In WES feedback, SST cooling over the western North Pacific suppresses in situ convective heating, exciting a westward-propagating Rossby wave to form the sub-high over the Philippine seas. The intensified northeast trade winds at the eastern edge of the sub-high amplifies the initial SST cooling via evaporation and wind stirring. It has been suggested that the Indian Ocean warming and sub-high intensification in summer occur when the initial disturbances that form the IPOC mode (e.g., tropical Northwest Pacific SST lowering) occur in the spring season. Thus we elected to analyse

the climate in spring. We have revised the manuscript to clarify these points. (Line 241−252 in the revised manuscript)

**Comment 11:** *Do you see a westward-propagating rossby wave in your result, which seems to be critical in the WES feedback explained above.*

**Response to comment 11:** As a Rossby response, we see an anomalously high pressure over the Philippine Sea. We have revised the manuscript to clarify this point. (Lines 247 and 254−256 in the revised manuscript)

**Comment 12:** *Please explain how spring climate is linked to summer insolation.*

**Response to comment 12:** Instead of solely responding to strong summer insolation, the spring climate may reflect the effects of increasing seasonal variations in insolation (e.g., weak winter insolation), suggesting that the East Asian summer monsoon can be influenced not only by summer insolation but also by variations in insolation distribution across other seasons. However, the mechanism linking insolation variation to SST variability is not yet completely understood and remains a challenge for future research. We have added explanations of these points to the revised text. (Lines 257−260 in the revised manuscript)

**Comment 13:** *this is related to a negative anomaly in the sea level pressure in northern Pacific (Fig.6a), but a intensification of the NPH is suggested in the rest of this section. Please explain.*

**Response to comment 13:** Thank you for your careful review of our manuscript. As mentioned in our responses to comments 7 and 8, we have replaced Fig. 6a with the correct version. The corrected Fig. 6a now shows the intensification of the NPH.

**Comment 14:** *Copernicus Publications requests depositing data that correspond to journal articles in reliable (public) data repositories, assigning digital object identifiers, and properly citing data sets as individual contributions.* see https://www.climate-of-the-past.net/policies/data_policy.html

**Response to comment 14:** The output data from the climate simulations conducted in this study (91 global climate simulation runs) occupy several terabytes of data in total and are therefore difficult to share publicly and free of charge through data repositories. However, we can consider providing the data individually upon request to the corresponding author. The area-averaged precipitation and temperature data used to create the time series and scatter plots will be made

available as a Supplement to this paper. These points have been specified in the Data Availability section of the revised manuscript.

**Comment 15:** *simulated? Isn't the insolation value obtained from Andre Berger's calculation?.*

**Comment 16:** *Explain how was this average done. and the anomaly is relative to present day?*

**Response to comments 15 and 16:** Using orbital parameters taken from Berger and Loutre (1999), we simulated the surface solar radiation in the model and averaged it over grid cells ranging from 20°N to 40°N. We have revised the text to clarify these points. The anomaly is relative to present day.

**Comment 17:** *at least the 100-kyr cycle in the proxy records could be related to glacial condition changes, not to eccentricity..*

**Response to comment 17:** The text has been revised as per your suggestion

**Comment 18:** *Please also explain the shaded curves on the top and to the right.*

**Response to comment 18:** The shaded curves show the normal distribution obtained from the mean and standard deviation of each data. The text has been revised to more clearly state this point.

**Comment 19:** *See my comment and modification for Fig 2's caption. please make similar modification for Fig.4's caption.*

**Response to comment 19:** The text has been revised as per your suggestion.

**Comment 20:** *See my comment and modification for Fig 2's caption. please make similar modification for Fig.4's caption.*

**Response to comment 20:** The text has been revised as per your suggestion.

**Comment 21:**

1. *Please plot precipitation change also over land, including significance level.*
2. *I would suggest to show the same figures also for the reference experiment.*
3. *why are the sea level pressure and surface wind anomalies not the same between Fig.6a and 6b although they are supposed to be the same variables?*

4. *The color bar is very confusing. In all three figures, I suggest to use blue for decrease and red for increase. For same variables (sst, wind stress), better to use the same scale.*

**Response to comment 21:** We appreciate the feedback.

1. We have replaced Fig 6a with the correct version (please see our responses to comments 7 and 8).
2. We have added a supplementary figure (Fig. S1) to show the output of reference run (0 ka climatology).
3. We have replaced Fig 6a with the correct version (please see our responses to comments 7 and 8).
4. The figures have been revised according to your suggestion. Brown is used to indicate a decrease and green to indicate an increase in precipitation (Fig. 6a), while blue represents a decrease and red an increase in SST (Fig. 6b and Fig. 6c). The figures now use the same scale for SST and wind.

Additionally, to address the difficulty in seeing some deviations (e.g., SST anomaly in MAM) due to the uniformity of the color bars, we have added a new figure plotted over a wider area to the supplement (Fig. S2).

**Comment 22:** *Figure 7?*

**Response to comment 22:** Thank you for the feedback. The Figure number has been revised accordingly.